# Enhanced Recovery after Surgery Rehabilitation Protocol in the Perioperative Period of Orthopedics: A Systematic Review

**DOI:** 10.3390/jpm13030421

**Published:** 2023-02-26

**Authors:** Jiasheng Tao, Zijian Yan, Guowen Bai, Hua Zhang, Jie Li

**Affiliations:** 1The First Clinical Medical College, Guangzhou University of Chinese Medicine, Number12, Jichang Road, Baiyun District, Guangzhou 510405, China; 2Hospital for First Affiliated Hospital of Guangzhou University of Chinese Medicine, 16th Jichang Road, Baiyun District, Guangzhou 510405, China

**Keywords:** enhanced recovery after surgery, orthopedics, surgery, rehabilitation, systematic review

## Abstract

Purpose: Enhanced recovery after surgery (ERAS) is a surgical rehabilitation protocol of increasing interest to clinicians in recent years, with the aim of faster and better recovery of patients after surgery. Our main focus in this review is to analyze the effectiveness of ERAS rehabilitation protocols in orthopedic surgery. By comparing the post-operative recovery of patients receiving the ERAS rehabilitation program with that of patients receiving the conventional rehabilitation program, we observed whether the patients who have received the ERAS rehabilitation program could recover better and faster, thereby achieving the aim of a shorter hospital stay and reducing the incidence of complications. Methods: We conducted the literature searches in PubMed, MEDLINE, Web of Science, Cochrane Reviews, EMBASE and other databases on clinical studies related to orthopedic surgery regarding the effectiveness of rehabilitation using ERAS rehabilitation protocols compared with conventional rehabilitation protocols. A systematic review was performed in accordance with the Preferred Reporting Items of Systematic Reviews and Meta-analysis (PRISMA) statement. If there was variability in the rehabilitation data of the patients between the two subgroups, it was considered that there was a difference in the rehabilitation effect of the ERAS rehabilitation protocol and the conventional rehabilitation protocol on the patients. Conclusion: The application of ERAS rehabilitation protocols can shorten patients’ hospital stay and reduce their expenses. In addition, patients with ERAS rehabilitation protocols will have fewer postoperative complications, while patients will have less postoperative pain than those with conventional rehabilitation, facilitating better postoperative recovery.

## 1. Introduction

Orthopedics as one of the important component departments of a general hospital, whether it is for patients with fractures, joint replacements, or spine surgery. The surgical trauma for the patient is often large and is prone to corresponding complications, which can have a definite impact on the patient’s psychological and physical function. Therefore, the rehabilitation of patients after orthopedic surgery is very important. Good rehabilitation can go a long way towards improving the patient’s prognosis and facilitating the corresponding functional recovery after surgery. It can also play an important role in maintaining the patient’s psychological well-being. If a rapid recovery plan can be developed for patients in the perioperative period from preoperative to postoperative, in order to reduce the length of hospitalization, better restore the patient’s limb function, and reduce postoperative complications, it can promote both orthopedic surgery and improve the patient’s treatment outcome and psychological recovery. It is of great clinical significance to promote the rehabilitation of patients after orthopedic surgery.

Enhanced Recovery After Surgery (ERAS) is a multimodal, interdisciplinary care improvement program designed to facilitate the recovery of patients who have undergone surgery during the perioperative period; the purpose is to shorten the length of hospital stay and reduce complications [1,2]. The ERAS rehabilitation program is a rehabilitation treatment measure that has been increasingly used in the perioperative period for patients in recent years, and its rehabilitation outcomes have gotten more widely recognized. Evidence-based standardization of the perioperative management of surgical patients through the implementation of ERAS programs to improve outcomes is the most important clinical aim.

The enhanced recovery after surgery (ERAS) program is increasingly used in orthopedic surgery. In an ERAS rehabilitation progr for orthopedic related surgery, the medical members involved usually consist of orthopedic specialists, nurses, anesthetists and rehabilitation staff. The ERAS rehabilitation program involves a series of components that combine to minimize stress and to facilitate the return of function: these include integrated pre-operative, intra-operative and post-operative management [3,4]. The preoperative management includes mainly educational programs, nutritional management, dietary management, sleep management and pain management; the intraoperative treatment includes four main aspects: selection of anesthesia, goal directed fluid therapy, temperature management and infection prevention [5]; the postoperative management includes five main aspects: pain management, rehydration management, drainage management, nausea and vomiting control and activity management [6,7,8]. The purpose of the perioperative ERAS rehabilitation program is to enhance the patient’s postoperative recovery, reduce complications, and shorten the length of hospital stay.

In order to comprehensively analyze the application and rehabilitation outcomes of ERAS rehabilitation programs in orthopedic surgery, we collected and compared the rehabilitation outcomes between patients with ERAS rehabilitation programs and those with conventional rehabilitation programs in all directions of orthopedic surgery in this review. The advantages of the ERAS rehabilitation programs over conventional rehabilitation programs were investigated. The aim of our study is to promote the better use of ERAS rehabilitation programs in orthopedic surgery and to improve patient outcomes and postoperative recovery.

## 2. Methods

### 2.1. Literature Search

We strictly followed the Preferred Reporting Items of Systematic Reviews and Meta-analysis (PRISMA) statement in conducting the relevant literature searches [9,10]. The databases searched included PubMed, MEDLINE, Web of Science, Cochrane Reviews, EMBASE. As there are no studies on postoperative ERAS in orthopedics before 1960, so the time frame for the search is from 1960 to 30 August 2022. Methods for including and excluding articles include reading the title and abstract of the article and excluding the articles that do not meet the criteria. Then we carefully read the specific content of the remaining articles, further excluded the articles that do not meet the standards, and finally determined the specific articles to be included.

### 2.2. Inclusion and Exclusion Criteria

We developed the following inclusion criteria based on the needs of the study: (1) published peer-reviewed reports of human studies in English; (2) publication dates between database creation and 30 August 2022; (3) complete clinical reports; and (4) clinical subgroups of patients who were rehabilitated using ERAS and conventional rehabilitation, respectively.

The following are shown as exclusion criteria: (1) reviews, hypotheses, technical articles or oral reports; (2) non-English articles; (3) patients who have previously undergone surgery; (4) cadaveric or animal studies; (5) no control group; or (6) the control group received ERAS rehabilitation therapy after surgery.

### 2.3. Data Extraction and Quality Assessment

We first included and excluded articles by reading the title and abstract sections of the articles, eliminating those that did not meet the requirements, and then carefully read the full text of the remaining articles. The full text was read to exclude articles that did not meet the inclusion criteria, thus identifying articles that met the inclusion criteria. After we identified the final included literature, we arranged for two researchers to conduct a quality assessment of the included articles. After completion of a risk assessment for quality and bias, the articles were subjected to data extraction, which included the first author, year of publication, study design, study period, indication for surgery, sample size, number of patients in each control group, gender, mean age, type of intervention (ERAS rehabilitation program or conventional rehabilitation program), length of hospital stay, complication profile and visual analogue scale profile.

In addition, after data extraction was completed, the quality within each study was assessed. We assessed the literature for quality by referring to the assessment manual provided by the Cochrane system [11]; the tool used for quality assessment was Revman 5.4. Risk items assessed included random sequence, allocation concealment, blinding, completeness of outcome data, risk of selective reporting bias, and other biases. Each item was rated as ‘low risk’, ‘high risk’ or ‘uncertain risk’. A lower risk of bias indicates a higher quality of the included literature.

### 2.4. Statistical Analysis

The statistical analysis software used in our study was Stata SE-64. The main outcome indicators were compared between patients with the postoperative ERAS rehabilitation program and patients with the postoperative conventional rehabilitation program in order to analyze the differences in the postoperative rehabilitation of patients with the two different rehabilitation programs. A 95% confidence interval was used for the outcome indicators for the dichotomous variables to maintain consistency of analysis. Mean differences (MD) were expressed as 95% confidence intervals (95% CI) associated with continuous variables. If studies reported only the median, range, and size of the trial, we used the data reported in the paper to calculate the mean and standard deviation [12]. In addition, we have analyzed the heterogeneity of the included articles by I^2^ statistics. If the I^2^ value was 0% it indicated no heterogeneity, while I^2^ < 25% indicated low heterogeneity, the I^2^ value of 25–50% moderate heterogeneity and I^2^ > 50% high heterogeneity [13].

After counting the data from the included articles, we analyzed the data. The statistical analysis methods included fixed effect model and random effect model. When I^2^ < 50% indicated little heterogeneity within the literature, a fixed effects model (FE) was chosen to analyze the data. When I^2^ > 50% indicated significant heterogeneity within the literature, a random effects model (RE) was chosen to analyze the data. The differences between ERAS rehabilitation and conventional rehabilitation outcome indicators were analyzed according to the resulting forest plots. Statistical significance is considered if the *p*-value is <0.05.

According to the description in the respective articles, the patients who had used ERAS rehabilitation in the perioperative period were included in an ERAS rehabilitation group and those who did not receive ERAS rehabilitation were grouped into a conventional rehabilitation group. Among the included studies, we summarized and aggregated the various data provided in the article, in which the more frequently reported outcome indicators were length of hospitalization, postoperative complications, and patient-related postoperative pain scores (VAS). Therefore, the primary outcome indicators in our current analysis are length of stay and complications, and the secondary outcome indicator is pain score. In the included articles, all reported postoperative complications were included in the outcome index of complications, such as postoperative gastrointestinal bleeding, infection of the surgical opening, urinary tract infection, respiratory tract infection, thrombosis and other postoperative complications. The length of stay in hospital and the presence of complications were used to analyze the recovery of patients after undergoing orthopedic surgery together with the length of recovery time. The VAS pain score allowed us to analyze the pain of orthopedic patients after surgery.

## 3. Results

### 3.1. Identification of Included Studies

After excluding the articles that did not meet the inclusion criteria, a total of 40 articles were included. Of these articles, 11 are for fracture-related procedures [14,15,16,17,18,19,20,21,22,23,24]. There are 14 articles on joint replacement-related surgery [25,26,27,28,29,30,31,32,33,34,35,36,37,38] and 15 articles on spine surgery [39,40,41,42,43,44,45,46,47,48,49,50,51,52,53]. The relevant literature search and exclusion process is shown in Figure 1.

### 3.2. Quality Assessment of Included Studies

Of the 40 clinical studies included, 2 randomized controlled trials (RCT), 31 retrospective case-control studies, and 7 prospective cohort studies were included. A total of 29,856 patients were included in all articles, of which a total of 10,991 patients underwent surgery related to fracture, 3607 patients underwent surgery relating to the spine, and 15,258 patients underwent joint replacement surgery. Table 1, Table 2 and Table 3 show specific article characteristics and ratings. The assessed risk of bias in the included articles is shown in Figure 2A,B.

### 3.3. Length of Hospitalization

For the outcome indicator of length of stay, 9 articles were reported for fracture surgery, 11 articles for joint replacement surgery, and 14 articles for spine-oriented surgery. Figure 3 shows the forest plot obtained after our analysis of this data. Figure 3A shows the forest plot for fracture surgery (RR: −3.09; 95% Cl: −3.98, −2.20; *p* < 0.001, I^2^ = 99.6%), Figure 3B shows the forest plot related to joint replacement (RR: −1.30; 95% Cl: −2.56, 0.36; *p* = 0.141 > 0.05, I^2^ = 99.7%), and Figure 3C shows the forest plot related to spine orientation surgery-related forest plot (RR: −1.93; 95% Cl: −2.43, −1.42; *p* < 0.001, I^2^ = 91.9%). As for the analysis of the length of hospitalization, the length of hospital stay in patients with fracture or spinal surgery undergoing ERAS rehabilitation is found to be shorter than that of patients undergoing conventional rehabilitation, *p* < 0.05. There is no difference in the length of hospital stay for patients undergoing joint replacement regardless of the rehabilitation program.

### 3.4. Complications

The clinical outcome of complications occurring after surgery was reported in 9 articles for fracture surgery, 13 articles for joint replacement surgery, and 12 articles for spine-oriented surgery. Figure 4 shows the forest plot obtained from the analysis of this data set. Figure 4A shows the forest plot for fracture surgery (RR: 0.59; 95% Cl: 0.44, 0.79; *p* < 0.001, I^2^ = 39.6%), Figure 4B shows the forest plot related to joint replacement (RR: 0.42; 95% Cl: 0.33, 0.54; *p* < 0.001, I^2^ = 53.0%), and Figure 4C shows the forest plot related to spine-oriented surgery (RR: 0.69; 95% Cl: 0.51, 0.94; *p* = 0.017 < 0.05, I^2^ = 46.9%). Regardless of the type of surgery (fracture, spine, joint), the postoperative complications were significantly less in patients receiving ERAS rehabilitation than in patients receiving conventional rehabilitation, *p* < 0.05. The use of ERAS rehabilitation can significantly reduce the occurrence of postoperative complications.

### 3.5. Visual Analogue Scale (VAS)

There are relatively few data on this indicator of VAS pain score, and only data related to joint replacement and spine surgery are summarized. A total of 6 of these articles were reported for joint replacement surgery and a total of 6 articles were reported for spine-oriented surgery. Figure 5 shows the forest plot we obtained after analyzing this data. Figure 5A shows the forest plot related to joint replacement (RR: −1.17; 95% Cl: −1.82, −0.52; *p* < 0.001, I^2^ = 97.7%) and Figure 5B shows the forest plot related to spine-oriented surgery (RR: −0.91; 95% Cl: −1.17, −0.65; *p* < 0.001, I^2^ = 67.2%). After joint replacement or spine-related surgery, the patients with ERAS rehabilitation have significantly lower pain scores than those in the conventional rehabilitation group.

## 4. Discussion

According to the forest plots we obtained and the results of the analysis, there was no difference in the length of stay in patients who underwent joint replacement, either with the ERAS rehabilitation protocol or with the conventional rehabilitation protocol (*p* > 0.05). The LOS was approximately the same in both rehabilitation groups. However, there was a significant difference in postoperative complications between the two rehabilitation groups of patients in joint replacement surgery (*p* < 0.001). There was also significant variability in postoperative pain among arthroplasty patients who had received different rehabilitation protocols (*p* < 0.05). These results suggest that the patients who had received the ERAS rehabilitation program experienced less postoperative pain and complications than those who had received conventional rehabilitation. The use of the ERAS rehabilitation program can help patients to reduce the incidence of post-operative complications and reduce their pain and discomfort. This can go a long way in promoting better post-operative recovery for patients. There was no significant variability in the length of hospital stay for patients in the joint replacement group, which may be related to the patient’s age, underlying disease, psychological status, and intraoperative blood loss, which can lead to a longer postoperative hospital stay [54,55,56].

Overall, there were significant differences in outcome indicators, except for the length of hospital stay in joint replacement. In patients who had undergone fracture surgery and spine surgery, there was a significant difference in all outcome indicators between the ERAS rehabilitation group and patients in the conventional rehabilitation group (*p* < 0.05). The analysis results for patients who had undergone fracture surgery and spinal surgery show that the two different rehabilitation protocols had different effects on the patients’ post-operative recovery. The use of the ERAS rehabilitation program can help patients to recover better after surgery. These advantages are fully reflected in terms of length of stay, postoperative complications and VAS scores. The results of our analysis were consistent with those of some current studies. The ERAS rehabilitation program provides comprehensive rehabilitation management of patients before, during, and after surgery, and can improve postoperative outcomes, reduce complications, shorten hospital stays, and reduce the impact of postoperative pain in patients [4,57].

For the patients who had undergone joint replacement surgery, the length of stay in hospital was somewhat shorter for those with the ERAS rehabilitation program than for those in the conventional rehabilitation group. These results suggested that the ERAS rehabilitation can improve the patient’s post-operative condition and achieve the clinical requirements for discharge from hospital more quickly. These could show that the ERAS rehabilitation program can help patients recover better and faster, and thus return to normal life and work more quickly. For the patients in the ERAS rehabilitation group and the conventional rehabilitation group, there was a significant difference in post-operative complications and pain between the two. It could be that the two methods of rehabilitation have different factors affecting postoperative complications and pain,, or it may be that there were differences in the specific implementation process in the clinical setting which has an impact on the outcome. For example, the differences in the clinician’s surgical experience and technique, or differences in the specific pain relief protocols and pain medication used, may also have an impact on post-operative complications and pain. The impact of an ERAS rehabilitation program on complications, pain and even function after arthroplasty is a question that clinicians will need to consider and explore in the future, and deserves further evidence from more clinical trials.

In addition, as a multimodal management, the ERAS rehabilitation program can reduce the length of hospitalization of patients while also reducing the cost of patients’ expenses and reducing medical stress [58]. The results of related studies have also shown that the application of ERAS rehabilitation protocols not only reduces postoperative complications, but also decreases mortality and rehospitalization rates, improves joint and limb motility and neurological function, promotes muscle strength recovery, and improves patient prognosis more significantly [24,59]. For patients who had undergo the ERAS rehabilitation program, the pain symptoms that occur after surgery were much less severe. It can reduce the use of pain medication to some extent, thereby reducing some of the adverse effects associated with pain medication. In particular, reducing the use of addictive drugs such as opioid painkillers can greatly reduce some of the adverse reactions and complications caused by these drugs after surgery. This is of great clinical significance to the rapid and good recovery of patients after surgery [60,61,62]. To improve the postoperative status of orthopedic patients by reducing postoperative complications and pain is an aspect that clinicians need to pay attention to in the perioperative period. This can not only improve the overall therapeutic effect of patients, but also improve patients’ trust in doctors and form a habit of good medical compliance, which can play a positive role in promoting doctors’ clinical work. Therefore, the good application of the ERAS rehabilitation program may be a direction that doctors need to consider more in future clinical work.

Of course, there are still some concerns and controversies about some of the measures in the current ERAS rehabilitation program. For example, the ERAS Society often advocates a reduction in the use of anti-inflammatory analgesics along with early parenteral nutrition, which may increase the risk of postoperative inflammation. In terms of preventing post-operative thrombosis, the ERAS Society recommended that adequate thromboprophylaxis should be provided during the patient’s post-operative recovery period. Thromboprophylaxis can be gradually reduced and discontinued after the patient has fully resumed activity [63]. However, as some patients consider thromboprophylaxis to be non-essential, pharmacological thromboprophylaxis may increase the risk of bleeding. Adequate thromboprophylaxis may therefore partly lead to patient refusal [64]. Therefore, a well-developed ERAS rehabilitation program that allows patients to receive more comprehensive and complete management in the perioperative period, such as reducing the risk of postoperative thrombosis through preoperative intervention, may be more acceptable to patients. In addition, the ERAS Society strongly recommended the use of balanced crystalloids and avoidance of 0.9% saline, which may increase the incidence of postoperative hyperchloremia and hypotension [63]. For the above mentioned possible accompanying risks, potential adverse reactions can be avoided by adjusting the relevant steps. For example, the use of analgesic and anti-inflammatory drugs and infusion protocols can be adapted to the patient’s condition to achieve a better postoperative recovery while reducing the associated risks.

Based on our comprehensive analysis of postoperative patients in all directions of orthopedics, comprehensive rehabilitation management of patients by using ERAS in the perioperative period can provide clear benefits for patients. Compared to conventional rehabilitation, ERAS rehabilitation programs can significantly improve the overall postoperative condition of patients. Of course, in addition to improving length of stay, post-operative complications and pain, ERAS rehabilitation may also can improve other post-operative indicators for patients, such as the patient’s functional recovery and functional scores after surgery, the need for postoperative blood transfusions and the readmission rate after discharge. Improvements in all these indicators can make an important contribution to the patient’s postoperative recovery. More consideration can be given to the exploration of these indicators in future clinical studies on ERAS, together with a more comprehensive analysis of the benefits and drawbacks of ERAS rehabilitation programs for patients, thus promoting better clinical application of ERAS rehabilitation programs.

## 5. Limitations

Of course, there are some limitations to the current review. First, the vast majority of the included clinical research articles were retrospective, with fewer randomized controlled studies, which may have confounded the results. Second, the specific measures of ERAS rehabilitation protocols used by individual departments or hospitals vary to some extent and cannot be completely standardized, which may also cause bias in the final outcome.

## 6. Conclusions

In the perioperative period in all directions of orthopedics, the application of ERAS rehabilitation protocols can shorten the length of stay and reduce the cost of patients’ expenses compared to conventional rehabilitation protocols. In addition, patients benefiting from the ERAS rehabilitation program will have fewer postoperative complications, and patients will have less postoperative pain than those with conventional rehabilitation, which is more conducive to better postoperative recovery. Therefore, ERAS rehabilitation protocols deserve more consideration in the perioperative period for orthopedic patients.

## Figures and Tables

**Figure 1 jpm-13-00421-f001:**
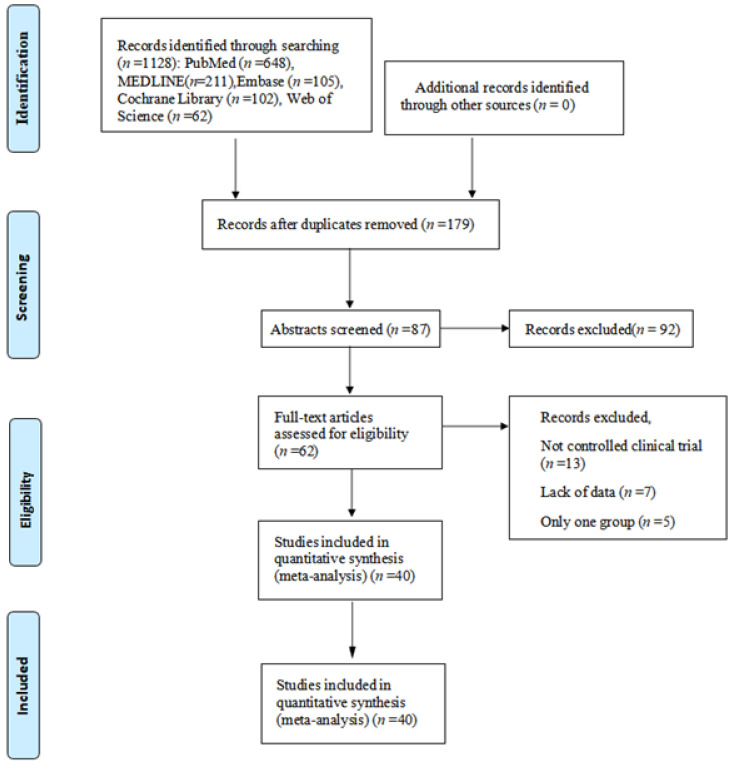
The inclusion process of the literature search.

**Figure 2 jpm-13-00421-f002:**
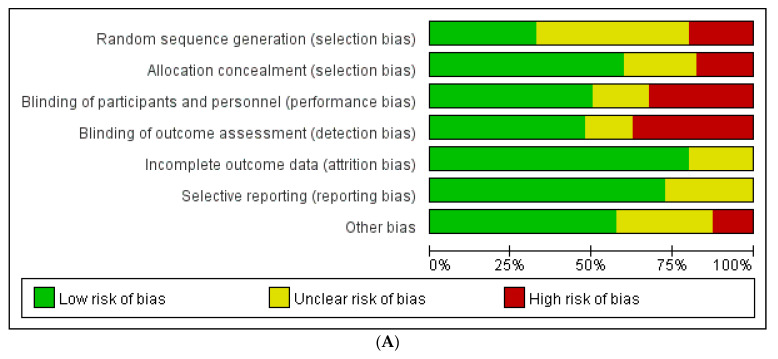
(**A**): Risk of bias assessment in studies. (**B**): Risk of bias assessment in each study [14,15,16,17,18,19,20,21,22,23,24,25,26,27,28,29,30,31,32,33,34,35,36,37,38,39,40,41,42,43,44,45,46,47,48,49,50,51,52,53].

**Figure 3 jpm-13-00421-f003:**
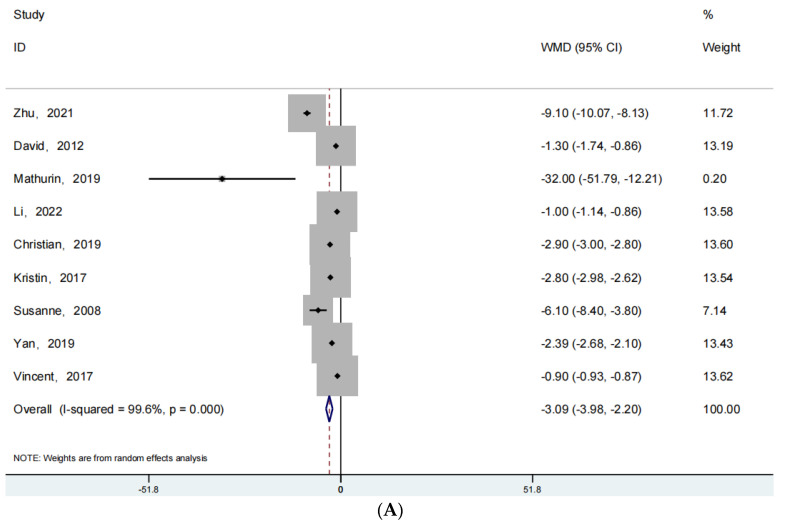
(**A**): The forest plot for fracture surgery of length of hospitalization. (**B**): The forest plot for joint replacement surgery of length of hospitalization. (**C**):The forest plot for spine surgery of length of hospitalization [25,26,27,28,29,30,31,32,33,34,35,36,37,38].

**Figure 4 jpm-13-00421-f004:**
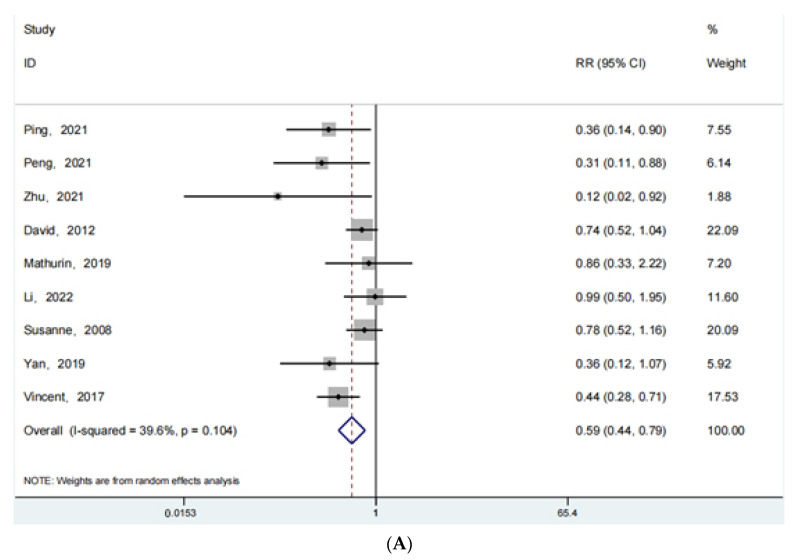
(**A**): The forest plot for fracture surgery of complications. (**B**): The forest plot for joint replacement surgery of complications. (**C**):The forest plot for spine surgery of complications [39,40,41,42,43,44,45,46,47,48,49,50,51,52,53].

**Figure 5 jpm-13-00421-f005:**
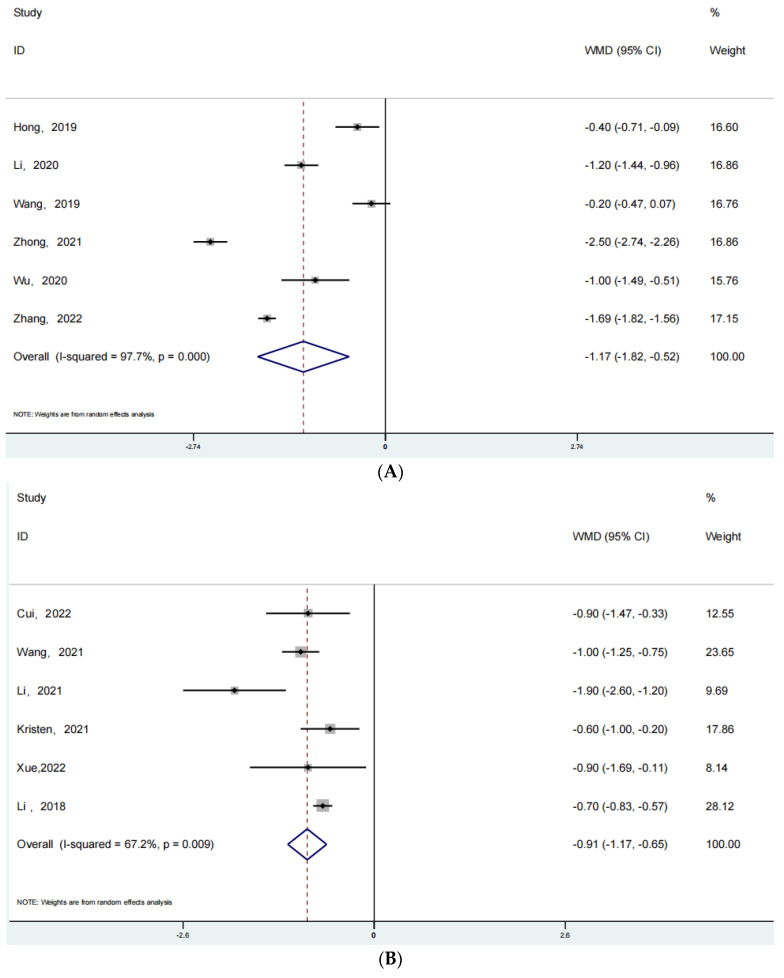
(**A**): The forest plot for joint replacement surgery of VAS. (**B**): The forest plot for spine surgery of VAS [39,40,41,42,43,44,45,46,47,48,49,50,51,52,53].

**Table 1 jpm-13-00421-t001:** The basic characteristics of the included studies of fracture.

Study (Ref.)	Type of Study	Number of Participants		Age (Years) (Mean ± SD)		Intervention		Quality of the Literature	Outcomes
		Trial	Control	Trial	Control	Trial	Control		
Ping, 2021 [11]	Retrospective	40	40	69.2	68.7	ERAS	CR	6	C
Peng, 2021 [12]	Retrospective	51	51	76.8	75.9	ERAS	CR	6	C
Zhu, 2021 [13]	Retrospective	92	98	78.1	77.3	ERAS	CR	7	LOS, C
David, 2012 [14]	Retrospective	117	115	82.5	82.7	ERAS	CR	7	LOS, C
Mathurin, 2019 [15]	Prospective	27	27	84.5	85.0	ERAS	CR	8	LOS, C
Li, 2022 [16]	Prospective	285	361	46.9	50.3	ERAS	CR	7	LOS, C
Christian, 2019 [17]	Retrospective	1140	1090	79.6	79.7	ERAS	CR	6	LOS, C
Kristin, 2017 [18]	Retrospective	1032	788	83.1	83.1	ERAS	CR	5	LOS
Susanne, 2008 [19]	Retrospective	178	357	83.9	84.2	ERAS	CR	6	LOS, C
Yan, 2019 [20]	Prospective	50	50	77.8	78.3	ERAS	CR	7	LOS, C
Vincent, 2017 [21]	Retrospective	2514	2488	63.2	62.1	ERAS	CR	6	LOS, C

ERAS: Enhanced recovery after surgery; CR: Conventional Rehabilitation; C: Complication; LOS: Length of hospitalization.

**Table 2 jpm-13-00421-t002:** The basic characteristics of the included studies of joint replacement.

Study (Ref.)	Type of Study	Number of Participants		Age (Years) (Mean ± SD)		Intervention		Quality of the Literature	Outcomes
		Trial	Control	Trial	Control	Trial	Control		
Hong, 2019 [22]	Prospective	106	141	74.2	75.4	ERAS	CR	7	LOS, C, VAS
Cao, 2021 [23]	Retrospective	183	178	66.1	66.0	ERAS	CR	6	LOS, C
Li, 2020 [24]	Retrospective	86	82	4.0	4.2	ERAS	CR	6	LOS, C, VAS
Wei, 2021 [25]	Retrospective	60	60	65.8	65.6	ERAS	CR	6	LOS, C
Wang, 2019 [26]	RCT	59	59	63.0	64.1	ERAS	CR	9	LOS, C, VAS
Liao, 2022 [27]	Retrospective	40	40	64.8	65.3	ERAS	CR	8	C
Zhong, 2021 [28]	Prospective	180	168	64.0	65.0	ERAS	CR	5	LOS, C, VAS
Marinus, 2016 [29]	Retrospective	100	100	66.7	65.4	ERAS	CR	6	LOS, C
Wu, 2020 [30]	RCT	16	16	35.6	31.7	ERAS	CR	9	LOS, C, VAS
Wang, 2020 [31]	Retrospective	91	105	66.7	67.0	ERAS	CR	7	LOS, C
Xu, 2019 [32]	Retrospective	1724	4923	66.6	66.7	ERAS	CR	5	LOS, C, VAS
Zhang, 2022 [33]	Retrospective	50	50	68.0	70.0	ERAS	CR	7	C, VAS
Collett, 2021 [34]	Retrospective	100	196	67.7	66.7	ERAS	CR	8	C
Hong, 2020 [35]	Prospective	1423	4902	66.7	66.6	ERAS	CR	7	LOS

ERAS: Enhanced recovery after surgery; CR: Conventional Rehabilitation; C: Complication; LOS: Length of hospitalization; VAS: Visual Analogue Scale.

**Table 3 jpm-13-00421-t003:** The basic characteristics of the included studies of spine.

Study (Ref.)	Type of Study	Number of Participants		Age (Years) (Mean ± SD)		Intervention		Quality of the Literature	Outcomes
		Trial	Control	Trial	Control	Trial	Control		
Cui, 2022 [36]	Prospective	46	54	79.1	79.2	ERAS	CR	5	LOS, C, VAS
Adrien, 2021 [37]	Retrospective	44	44	55.1	55	ERAS	CR	7	LOS
Li, 2020 [38]	Retrospective	91	169	69.6	73.3	ERAS	CR	6	LOS, C
Wang, 2021 [39]	Retrospective	60	60	47.9	46.6	ERAS	CR	8	LOS, C, VAS
Gong, 2021 [40]	Retrospective	46	45	55.2	56.8	ERAS	CR	8	LOS, C
Bertrand, 2020 [41]	Retrospective	271	268	49.5	47.3	ERAS	CR	7	LOS, C
Wang, 2020 [42]	Retrospective	95	95	72.4	70.8	ERAS	CR	7	LOS, C
Li, 2021 [43]	Retrospective	60	67	73.6	74.3	ERAS	CR	6	LOS, C, VAS
Feng, 2019 [44]	Retrospective	44	30	61	59	ERAS	CR	6	LOS, C
Kristen, 2021 [45]	Retrospective	39	78	15.0	14.3	ERAS	CR	5	LOS, VAS
Zuo, 2021 [46]	Retrospective	84	95	71.3	71.6	ERAS	CR	6	LOS, C
Xue, 2022 [47]	Retrospective	70	73	53.2	52.1	ERAS	CR	6	LOS, C, VAS
Wang, 2022 [48]	Retrospective	530	530	65.0	64.2	ERAS	CR	7	C
Armagan, 2018 [49]	Retrospective	183	267	61.0	60.0	ERAS	CR	6	LOS
Li, 2018 [50]	Retrospective	114	110	58.5	56.9	ERAS	CR	7	LOS, C, VAS

ERAS: Enhanced recovery after surgery; CR: Conventional Rehabilitation; C: Complication; LOS: Length of hospitalization; VAS: Visual Analogue Scale.

## Data Availability

The articles and data are available on PubMed, MEDLINE, Web of Science, Cochrane Reviews, EMBASE, etc.

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
