# Peer review of "Enhanced Recovery after Surgery Rehabilitation Protocol in the Perioperative Period of Orthopedics: A Systematic Review"

_jpm, 2023, doi:10.3390/jpm13030421_

Round 1

Reviewer 1 Report

The authors performed a systematic review and meta analysis of ERAS in orthopedic surgery. They were systematic in their approach and are to be commended for abstracting meaningful information from the 40 studies they identified for inclusion. 

I have no major criticism but one comment. Regarding the lack of a statistically significant difference in the LOS for patients undergoing joint replacement surgery, LOS is but one indicator of the benefits of standardization of perioperative care that comes with an ERAS protocol. The authors would do well to note this and push future researchers/clinicians to look for and characterize these benefits (patient reported outcomes, opioid usage, functional outcomes, faster return to work/activities of daily living, etc).

Author Response

Dear Reviewer,

Thank you very much for your time involved in reviewing the manuscript and your very encouraging comments on the manuscript.
Thank you for your careful identification and kindly reminder of the lack of a statistically significant difference in LOS in patients undergoing joint replacement surgery. In response to your comments, we have added a discussion section and add a perspective on the possible implications for this condition. The added sections are shown below.

For the patients undergoing joint replacement surgery, the length of stay in hospital is shorter for those with the ERAS rehabilitation programme than for those in the conventional rehabilitation group. This suggests that undergoing ERAS rehabilitation can improve the patient's post-operative condition and achieve the clinical requirements for discharge from hospital more quickly. This goes some way to show that the ERAS rehabilitation programme can help patients recover better and faster, and thus return to normal life and work more quickly. However, there is no significant difference between patients with the two rehabilitation modalities in terms of postoperative complications and pain scores. For the patients in the ERAS rehabilitation group and the conventional rehabilitation group, there is no significant difference in post-operative complications and pain between the two. It may be that the difference between the two rehabilitation modalities in affecting post-operative complications and pain is not significant, or it may be that there ae differences in the specific implementation process in the clinical setting which has an impact on the outcome. For example, differences in the clinician's surgical experience and technique, or differences in the specific pain relief protocols and pain medication used, may have an impact on post-operative complications and pain. The impact of an ERAS rehabilitation programme on complications, pain and even function after arthroplasty is a question that clinicians will need to consider and explore in the future, and deserves further evidence from more clinical trials.

Reviewer 2 Report

Line 15: ...we observed whether patients who were receiving...

Line 27: protocols had fewer postoperative complications, including less postoperative pain...

Line 38: What is "in all directions?" Please revise.

Line 58: I have not heard of "bleeding control" as part of ERAS.  I have heard of goal directed fluid therapy, which is not mentioned.

Line 60: How is anesthesia management a part of postoperative recovery?

Line 75, 80: When was the database created?

Line 85: What is meant by the term bioaugmentation?

Line 92: Can you describe the abbreviation NOS?

Line 101: relevant articles were initially obtained.

Line 179: "The length of hospital stay....group." This sentence is redundant since its already stated in the previous sentence.

Line 180: There was significant difference, so which group had more which group had less postop complications is important to state here.

Line 184: What are "better postoperative complications?"

Line 215: the comment about postop inflammation needs a reference.

Line 217: Why would patients refuse thromboprophylaxis?

Line 218: Change balanced crystal to crystalloids.

Author Response

Dear Reviewer,

Thank you very much for your time involved in reviewing the manuscript and your very encouraging comments on the manuscript.
Thank you for your careful identification and kindly reminder.           All the points you proposed to modify the article have been modified, and the modified parts have been marked in red.

The description of intraoperative bleeding control is not accurate and has been changed to goal directed fluid therapy.  Anesthesia management does not belong to the postoperative part, but pain management, the article has been revised.  The NOS Scale is an abbreviation of the Newcastle-Ottawa Scale.  The references to inflammation have been added.

There has also been increased discussion about which group has fewer postoperative complications, the additions are as follows:For the patients in the ERAS rehabilitation group and the conventional rehabilitation group, there is significant difference in post-operative complications and pain between the two.    It could be that the two methods of rehabilitation have different factors affecting postoperative complications and pain, or it may be that there ae differences in the specific implementation process in the clinical setting which has an impact on the outcome.    For example, differences in the clinician's surgical experience and technique, or differences in the specific pain relief protocols and pain medication used, may have an impact on post-operative complications and pain.    The impact of an ERAS rehabilitation programme on complications, pain and even function after arthroplasty is a question that clinicians will need to consider and explore in the future, and deserves further evidence from more clinical trials.

For some patients who refuse to take adequate preventive measures against thrombus after surgery, the explanations have been added and corresponding references have been made.       The additions are as follows: In terms of preventing post-operative thrombosis, the ERAS Society recommends that adequate thromboprophylaxis be provided during the patient's post-operative recovery period.      Thromboprophylaxis can be gradually reduced and discontinued after the patient has fully resumed activity [61].      However, as some patients consider thromboprophylaxis to be non-essential, pharmacological thromboprophylaxis may increase the risk of bleeding.      Adequate thromboprophylaxis may therefore partly lead to patient refusal [62].

Reviewer 3 Report

Thank you for the opportunity to review this paper concerning ERAS in orthopedic surgery.

This paper could be improved by:

1. A read-through for English language and general edits. Line 96 (repeated use of "II") Line 88 ('we developed two researchers") Line 217 ("cause patients to refuse."), Lines 182-183. What are spacers? (Line 94). Should the title be "... A Systematic Review" instead of "...A System Review"?

2. Adding the statistical methodology to the Methods section

3. Complications were assessed as an outcome, but how did the authors define complications? Was it a yes/no from the reviewed study? Was it any complications or clinically relevant complications? This needs to be defined in the methods

4. Explain how the authors determined a study was "ERAS". Was it just because the study said it was, or was compliance assessed? What is the definition of 'conventional rehabilitation'? More clarity on how these groups were chosen and defined is needed.

5. The reference list stops at 61, but the paper contains references up to 80

6. The two major orthopedic ERAS studies were not cited in this paper. See https://erassociety.org/guidelines/#filter=.orthopaedic for these papers. Also, consider literature from Wainwright, TW

7. The introduction and background do not cite the prevailing orthopedic literature, instead choosing cardiac ERAS literature as support. Consider revising this section to be more focused on orthopedic topics. See #6

8. Several systematic reviews have been conducted for orthopedic ERAS. How is this study different? 

Author Response

Dear Reviewer,

Thank you very much for your time involved in reviewing the manuscript and your very encouraging comments on the manuscript.
1.    Thank you for your careful identification and kindly reminder.                All the points you proposed to modify the article have been modified, and the modified parts have been marked in red.

2.    The statistical analysis section has been added and marked in red in the article, as shown below: The statistical analysis software used in our study is Stata SE-64.    The main outcome indicators are compared between patients with the postoperative ERAS rehabilitation programme and patients with the postoperative conventional rehabilitation programme in order to analyse the differences in the postoperative rehabilitation of patients with the two different rehabilitation programmes.    A 95% confidence interval is used for the outcome indicators for the dichotomous variables to maintain consistency of analysis.    Mean differences (MD) were expressed as 95% confidence intervals (95% CI) associated with continuous variables.    If studies reported only the median, range, and size of the trial, We use the data reported in the paper to calculate the mean and standard deviation.    In addition, we have analysed the heterogeneity of the included articles by I2 statistics.    If the I2 value is 0% it indicated no heterogeneity, I2 < 25% is low heterogeneity, the I2 value is 25%-50% moderate heterogeneity and I2 > 50% is high heterogeneity.    After counting the data from the included articles, we then analyse the data by using Stata SE-64.    The differences between ERAS rehabilitation and conventional rehabilitation outcome indicators are analysed according to the resulting forest plots.    Statistical significance is considered if the p-value is <0.05.

3.   The definition of complications has been added in the article: In the included articles, all reported postoperative complications were included in the outcome index of complications.   Such as postoperative gastrointestinal bleeding, infection of the surgical opening, urinary tract infection, respiratory tract infection, thrombosis and other postoperative complications.

4.  Information on how to define ERAS rehabilitation and conventional rehabilitation has been added to the article: According to the description in the article, the patients who used ERAS rehabilitation in the perioperative period are grouped into the ERAS rehabilitation group and those who did not receive ERAS rehabilitation are grouped into the conventional rehabilitation group according to the description in the article.

5.  Thanks for your careful reading of the references and kind reminder, and the parts that need to be modified have been modified.  Thanks for the recommendation of two major orthopedic ERAS studies, which have been cited in the article.  Your valuable comments make our article more complete.

6.  Compared with several published articles on the systematic review of ERAS in orthopedics, our current systematic review summarizes relevant ERAS studies in various directions of orthopedics.  In addition, our systematic review has also supplemented the directions not analyzed before.  Therefore, our systematic review can more comprehensively evaluate the impact of ERAS on postoperative rehabilitation of orthopaedic patients.

Round 2

Reviewer 2 Report

The paper still requires review by someone to correct English grammar.

Author Response

Dear Reviewer,

Thank you very much for your time involved in reviewing the manuscript and your very encouraging comments on the manuscript. The grammar of the article has been revised and the revisions have been marked in red.

Reviewer 3 Report

Thank you for adding a methods section, but this could still use some work. The authors mention the statistical program twice (Stata), which is redundant within the paragraph itself. Then, on line 164, it is mentioned a third time. That, and mentioning the p-value cutoffs again. Methods do not go in the Results section.

The methods section also does not contain the type or description of the model the authors used for analysis. Please explicitly state or describe which type of model or statistical test was used to generate the results.

Generally speaking, methods do not go in the Results section so I strongly recommend re-formatting lines 137 - 170 to be in the Methods section and not Results.

The Introduction and Discussion sections are better with this revision.

Author Response

Dear Reviewer,

Thank you very much for your time involved in reviewing the manuscript and your very encouraging comments on the manuscript. According to your suggestion, we have modified the methods and results as required, and the revisions have been marked in red in the manuscript.